SCIENCE FORUM

# A survey of research quality in core facilities

**Abstract** Core facilities are an effective way of making expensive experimental equipment available to a large number of researchers, and are thus well placed to contribute to efforts to promote good research practices. Here we report the results of a survey that asked core facilities in Europe about their approaches to the promotion of good research practices, and about their interactions with users from the first contact to the publication of the results. Based on 253 responses we identified four ways that good research practices could be encouraged: (i) motivating users to follow the advice and procedures for best research practice; (ii) providing clear guidance on data-management practices; (iii) improving communication along the whole research process; and (iv) clearly defining the responsibilities of each party.

**ISABELLE C KOS-BRAUN\*, BJÖRN GERLACH AND CLAUDIA PITZER\***

**\*For correspondence:** isabelle.
kos@pharma.uni-heidelberg.de
(ICK-B); Claudia.pitzer@pharma.
uni-heidelberg.de (CP)

**Competing interests:** The
authors declare that no
competing interests exist.

**Reviewing editor:** Peter
Rodgers, eLife, United Kingdom

## Introduction

Concerns about reproducibility in various areas of research have been growing for more than a decade (*Eisner, 2018*; *Ioannidis, 2005*; *Ioannidis et al., 2014*; *Prinz et al., 2011*). Possible causes for a lack of reproducibility include selective reporting, the pressure to publish, and the need for better training in the design and analysis of experiments (*Baker, 2016*; *Smaldino and McElreath, 2016*), and the scientific community has developed various guidelines to promote rigorous and transparent research practice (*Bespalov et al., 2020*; *Dirnagl et al., 2018*; *Freedman et al., 2017*; *Munafò et al., 2017*; *Nosek et al., 2015*; *Wilkinson et al., 2016*).

Core facilities have a central position in many areas of research in the life sciences because: (i) they provide access to state-of-the-art equipment and advanced skills in a cost- and time-effective way; (ii) they develop new technologies and transfer their technical and research expertise to the numerous scientists; (iii) they connect institutions and foster collaborations and interdisciplinary research (*Meder et al., 2016*). Core facilities also generate a substantial fraction of the scientific data at some institutions, thereby offering protection against bias in the design and analysis of experiments, and supporting transparency, rigor and reproducibility. Core facilities can also disseminate good laboratory practices and train early-career researchers in a way that has a lasting impact.

The Association of Biomolecular Resource Facilities (ABRF) surveyed over 200 core facilities to assess how they implemented guidelines from the US National Institutes of Health (NIH) on scientific rigor and reproducibility and whether these guidelines influenced services and technology development (*Knudtson et al., 2019*). The survey revealed that only about half of the facilities were fully aware of the NIH guidelines existence. The main factors and challenges affecting the rigor and reproducibility were "the lack of training, mentorship, expertise, or oversight", "poor sample quality", "inadequate standardization of protocols or guidelines, and data analysis", "poor experimental design" and "time pressure". In addition, the lack of interest from customers and the lack of authority was considered a hindrance to reproducible research. The most frequent tools used by facilities to improve rigor were quality control and standard operation procedures.

While the ABRF survey focused on the implementation of NIH guidelines on rigor and reproducibility, we decided to conduct a survey that assesses the general status quo regarding good

research practices at core facilities. We aimed to analyze strengths and weaknesses to identify the strategic improvements that could maximize rigor and reproducibility. Our survey addresses in detail the whole process from the first contact between the facility and the user to the publication of results, step by step. Furthermore, it distinguishes between full-service and self-service facilities to better account for their different operating modes. This allowed us to reveal the following additional aspects not included the ABRF survey. Among the problems affecting the research quality in core facilities are difficulties in the communication with their users, insufficient management systems (at all levels), and the lack of clear definition of who is actually responsible for the quality of data produced at the facility.

## Results

Our survey was sent to the leaders of 1000 core facilities in different fields of the life sciences in Europe. When ranked by types of facility, microscopy and FACS facilities were top, followed by genomics and proteomics. In total, we received 253 complete forms from over 30 types of facility, which differ in the techniques and expertise they offer (*Figure 1—figure supplement 1*). They also vary in the number of employees and users, and in the amount of data generated (*Figure 1—figure supplement 2*).

### Full service versus self-service facilities

Core facilities can be classified in three distinct groups depending on who performs the experiments at the facility: (a) facility staff; (b) external researchers or users; (c) facility staff and users. We call *full-service* facilities those offering an "all-inclusive service", where facility staff execute the experiment (with or without data analysis). *Self-service* facilities provide and maintain an infrastructure where users have access to equipment, training and expert advice. (Such facilities were called user laboratories in *Meder et al., 2016*). At *hybrid-service* facilities experiments are performed by facility staff and users. Most of the responses to our survey came from hybrid-service facilities, followed by full-service facilities and then self-service facilities (*Figure 1A*).

In addition to the distinction of who actually performs the experiments, the service range provided by the facilities varies as well, from a basic one consisting of processing the samples and sending back the data to an extended range

from experimental design to publication. However, we do not specifically distinguish between these options.

To assess the extent of quality procedures offered to the users during the whole research process from experimental design to publication, we asked twelve yes/no questions about quality practices (*Figure 1B*). The majority of facilities that responded offer training and guidance on experimental design, sample preparation, data analysis and help troubleshooting. They also offer support in writing relevant sections of publications. At the same time, the survey identified areas with potential for improvement, such as communication with users and management.

There are notable differences between the three operating modes (*Figure 1B*). Only about a quarter of self-service facilities keeps documentation of the experiments compared to >95% of full-service facilities. Similarly, storage of raw data is offered by only half as many self-service facilities (40%) compared to full-service facilities (82%). Furthermore, fewer self-service facilities provide standard experimental protocols because the users may bring their own. On the other hand, the full-service facilities tend to train their users less, they consider training less important and provide primarily theoretical training (*Figure 1B* and *Figure 1—figure supplement 3*). Only a half of full-service facilities provide guidance how to analyze raw data, because they analyze the data themselves (*Figure 1B* and *Figure 1—figure supplement 4*).

### Research quality: Lack of funding is the major obstacle to research quality

In order to identify what is critical for research quality, we asked core facilities an open-field text question to list what factors they consider the most important and which of these need to be improved at their facility. As can be seen from the *Figure 2A*, the most prominent ones are training and communicating with users, followed by having enough qualified staff, as well as up-to-date and well-maintained equipment. From these factors, hiring more staff and purchasing/maintaining equipment are the most in need of improvement. Interestingly, although not considered as important, the aspect listed second in need of improvement is management. Management was mentioned at many different

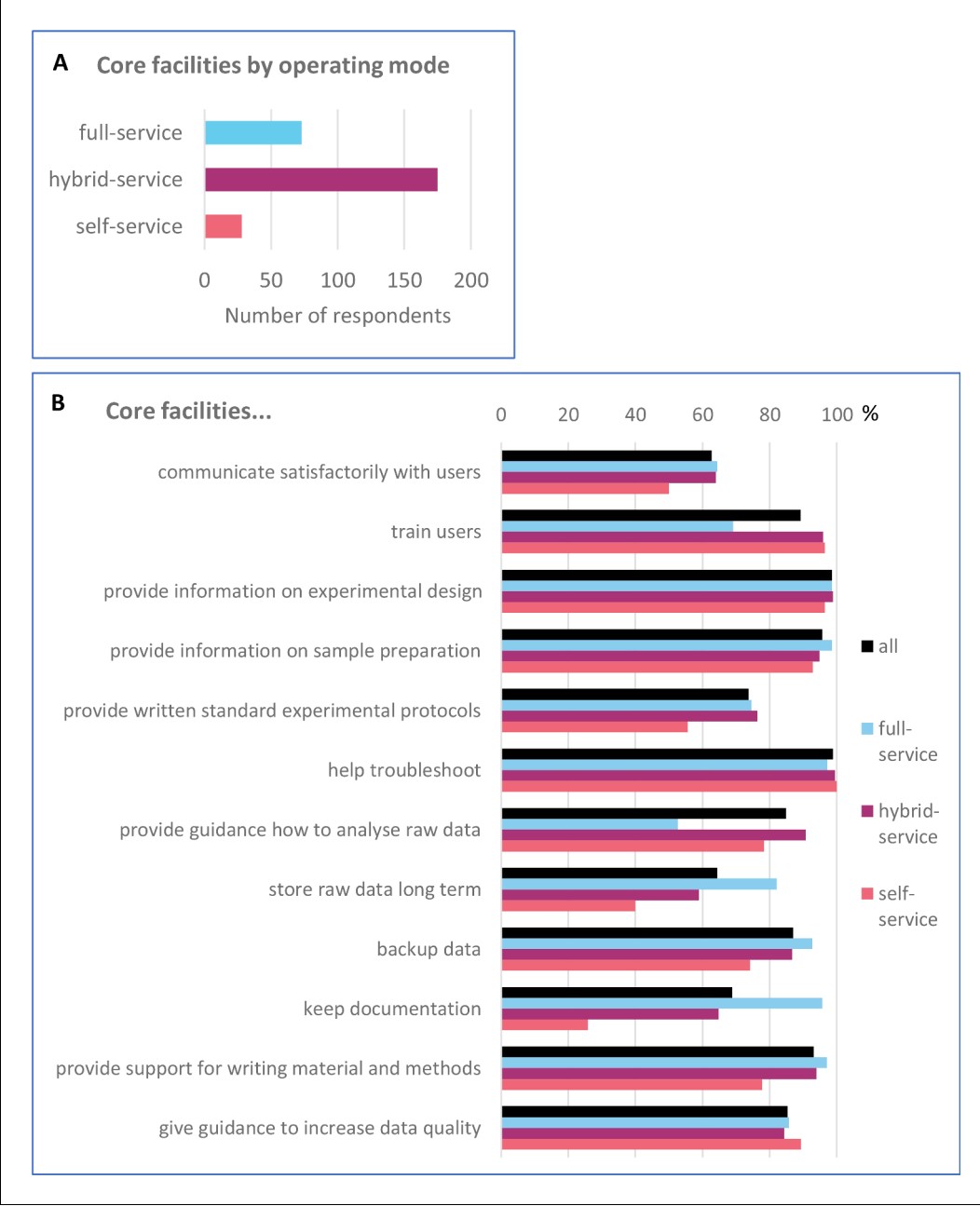

**Figure 1.** Comparison of core facilities by their operating mode and services offered. (**A**) Distribution of the surveyed core facilities (CFs) by their operating mode. (**B**) Fraction [%] of facilities providing different services along the research process. The overall fraction for all CFs, regardless of their operating mode, is depicted in black; different colours represent different operating modes.

The online version of this article includes the following figure supplement(s) for figure 1:

**Figure supplement 1.** Distribution of surveyed core facilities according to their type of technology.
**Figure supplement 2.** Characteristics of the surveyed core facilities.
**Figure supplement 3.** Opinion of core facilities on the importance of training and its aspects.
**Figure supplement 4.** Responsibility for raw data analysis.

levels: facility, projects, samples, data, IT infrastructure, documentation or automation (see the section on management).

When asked about the biggest challenges encountered by core facilities, the most frequently cited was the lack of funding, closely followed by the lack of staff (*Figure 2B*). The next

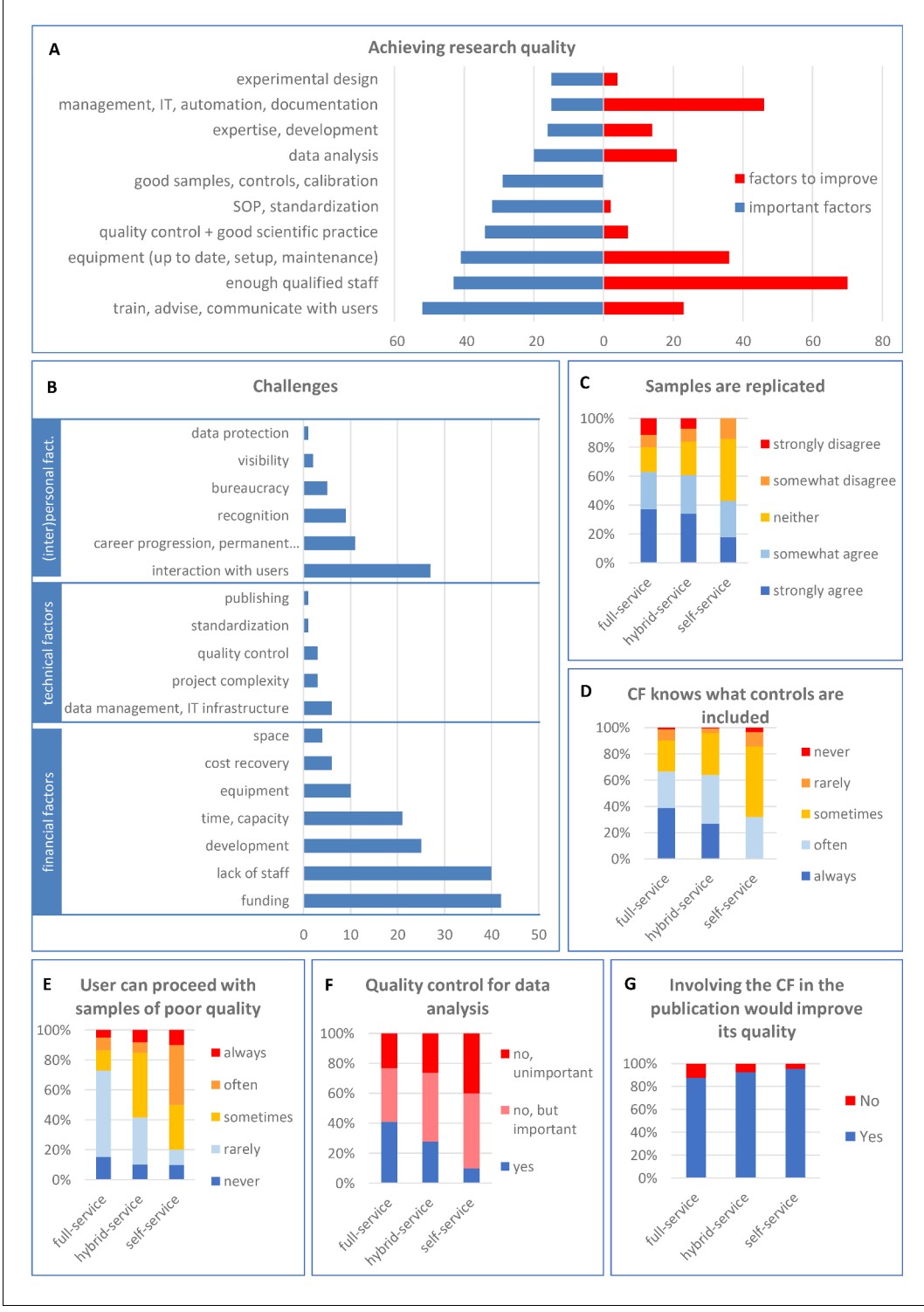

**Figure 2.** Research quality in core facilities: important factors, challenges and the current situation. (**A**) Facilities were asked for the most important factors for achieving research quality (in blue) and the aspects that need to be improved in their facility (in red; open-field question). (**B**) Challenges faced by core facilities, grouped in three categories (financial, technical and personal/interpersonal). The category "career progression" includes "permanent positions" and "motivation". (**C**) Facilities were asked if they agree or not that samples/experiments are replicated at their facility. Facilities were asked to rank on a 5-point scale whether they know what controls are included in experiments (**D**), and whether users are allowed to proceed with samples of poor quality (**E**). Facilities were also asked whether they have a quality control for data analysis (**F**) and, if not, how important such control

*Figure 2 continued on next page*

*Figure 2 continued*

would be. (G) Facilities were also asked if their involvement in manuscript preparation would improve the quality of published data.

The online version of this article includes the following figure supplement(s) for figure 2:

**Figure supplement 1.** Proportion of core facilities having sufficient funding and number of staff.

quoted were development (keeping the facility efficient and state-of-the-art), time and capacity. About half of the facilities that responded did not have enough funding and/or enough staff (*Figure 2—figure supplement 1*). The third ranked challenge concerns the interaction of facility staff with the users (see the section on communication).

### Research quality: Controlling quality from experimental design to publication

The process from experimental design to publication can be controlled at multiple checkpoints. As already shown in the *Figure 1B*, core facilities provide guidance along the whole process, from the experimental design, sample preparation, experimental protocol (standard operation procedures) to data analysis.

The experimental design defines the number of samples, which controls should be included, how the experiment will be performed and evaluated and how many replicates are necessary. We observe that experiments are often performed without replication, which is essential for good quality research (*Figure 2C*). When it comes to experimental controls, core facilities often do not know what controls were included (*Figure 2D*), even though experiments lacking appropriate controls cannot be meaningfully evaluated.

When asked if users could proceed with samples of poor quality, we found that 50% of self-service and 13% of full-service facilities often or always allow users to analyze samples of inadequate quality (*Figure 2E*). Regardless of the reason, whether it is due to the lack of controlling the sample quality or knowingly accepting such samples, it directly counteracts the efforts to achieve good quality research. Of course, samples of low quality can be justified in special cases. Introducing a sample quality checkpoint before starting the experiment is a simple measure that would clearly increase the quality of produced data (see discussion).

Regarding the data analysis, 40% of full-service facilities and only 10% of self-service facilities have mechanisms to ensure correct analysis and interpretation of raw data (*Figure 2F*).

Quality control of data analysis is usually performed by having the data checked by another staff member at the facility. It can also involve discussing the results with the user (or their principal investigator; PI), or using internal standards and quality control samples (data not shown). While the majority of core facilities do not control the quality of data analysis, most of them consider it important to have (*Figure 2F*).

The last opportunity for core facilities to check if the data they helped to produce was analyzed, interpreted and presented correctly is before publication. However, core facilities are often not even informed about the upcoming publication (see the section on communication below). The vast majority (91%) of all core facilities believe that if they were involved in the publication process it would improve the quality of published data (*Figure 2G*). This is most important for self-service facilities. The following quotations relate to the involvement of facility staff in the publication process:

- "[It] ensures correct understanding and an accurate account of what happened."
- "The users often lack the knowledge to use the correct controls or ways of display, without being aware that they are not following best-practice."
- "The core facility can ensure that the methods are detailed so that they can be replicated."

In conclusion, it is pertinent to introduce checkpoints to control the experimental design, sample quality, data analysis, and methods section and figures for publication. This seems particularly important in self-service facilities where more supervision would benefit the research quality.

### Management

Management is a very important factor for achieving research quality and many core facilities recognized the need to improve it (*Figure 2A*). There are many aspects of management, such as managing the budget, users, projects, samples and generated data.

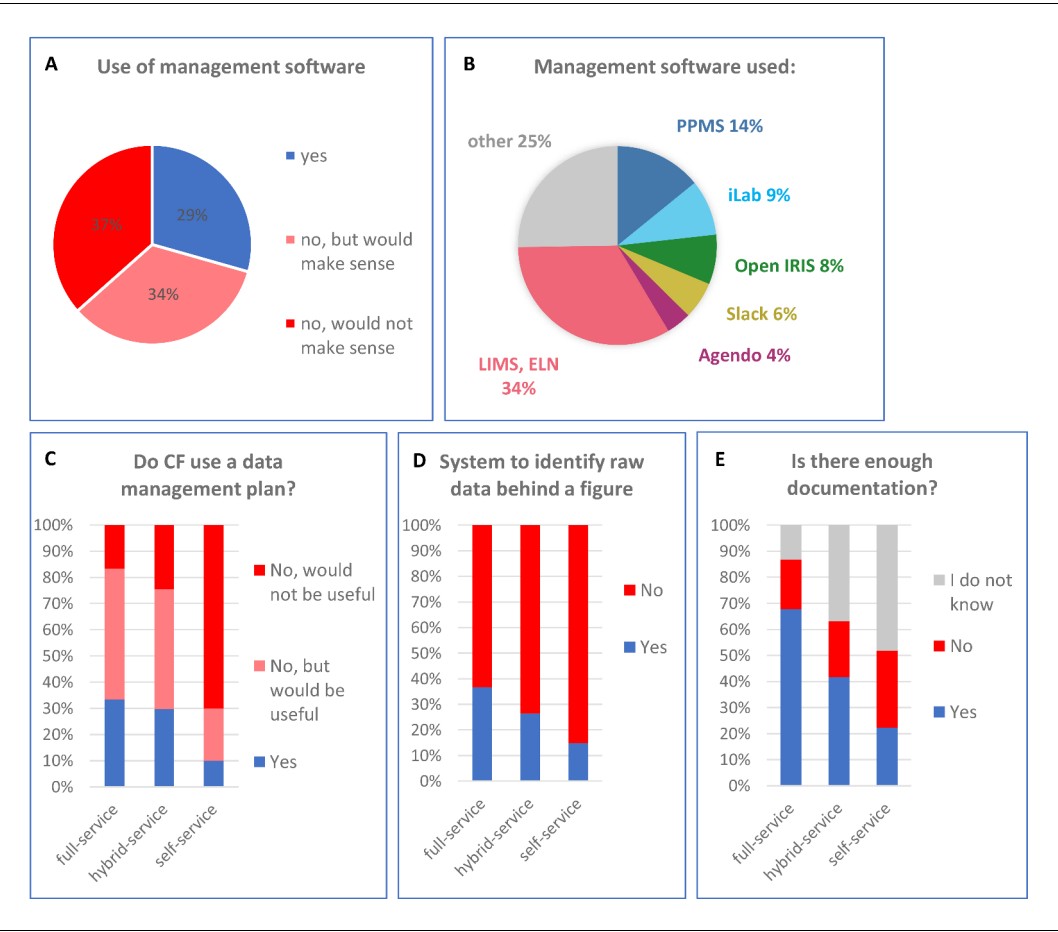

**Figure 3.** The use of management software and data management in core facilities. (A) Facilities were asked if they use management software and, if not, whether it would be useful. (B) Proportion of different management software used by core facilities. Software used by only one or two facilities is included under "Other". In a series of "yes" and "no" questions, facilities were asked if they use data management plans (C), have a system to identify the raw data behind a published figure (D), and have sufficient documentation (E). The results were normalized for operating mode.

The online version of this article includes the following figure supplement(s) for figure 3:

**Figure supplement 1.** Core facilities were asked in an open question which tools could be used to improve research quality.

**Figure supplement 2.** Comparison of two categories of management software used in core facilities showing the steps of the experimental process they cover.

**Figure supplement 3.** Management software used in core facilities by operating mode.

**Figure supplement 4.** Current situation in core facilities regarding different aspects of data management.

A management software is the tool most frequently used by facilities to achieve research quality (*Figure 3—figure supplement 1*). Overall, close to 30% of facilities use a management software and further 34% believe it would make sense to use one (*Figure 3A*).

Currently, the management software used in core facilities can be split into two categories (*Figure 3—figure supplement 2*). The first category, "core facility management software" mainly allow facilities to communicate with their users, book equipment and manage access rights, training, maintenance, technical issues and billing. It can also manage individual projects to a certain extent and keep records in the form of uploaded documents. Examples of such management software are PPMS from Stratocore, iLab from Agilent, Agendo or Open IRIS (open source). The second category is the "data management software". Electronic Lab Notebooks (ELN) allow the precise recording of scientific procedures from the experimental design

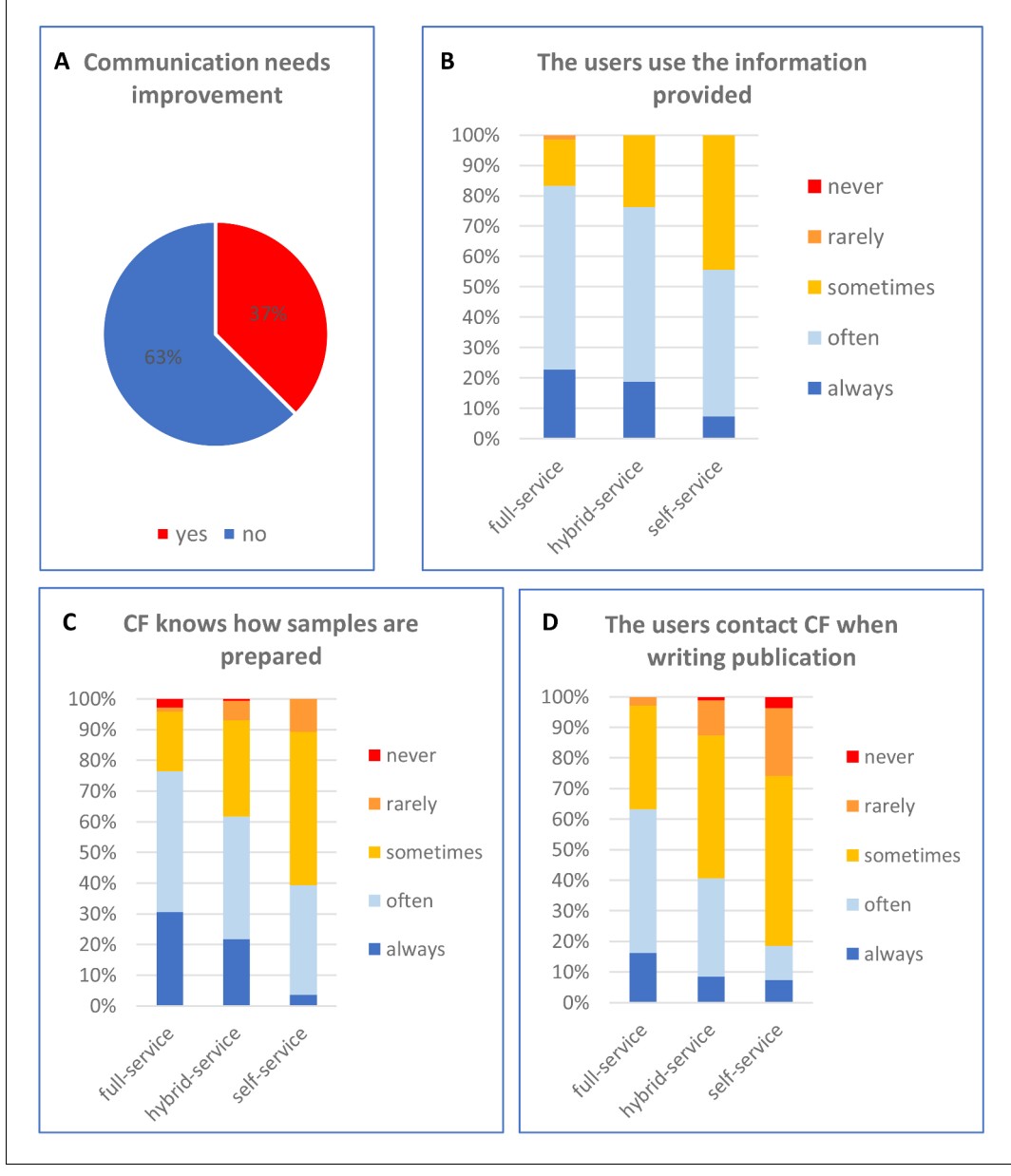

**Figure 4.** Current state of communication and interaction between core facilities and users. FFs were asked if the communication between facility staff and users needs to be improved (**A**), if users use the information provided by facility staff (**B**), facility staff know how samples have been prepared (**C**), and if the users contact the facility when they are writing a manuscript (**D**).

and sample preparation to the publication. It manages data acquisition, storage and analysis. This interconnected documentation ensures transparency and traceability. The Laboratory Information Management Systems (LIMS) are similar although they are often linked to one piece of equipment.

Our survey revealed that the software solutions used by core facilities are very heterogeneous (*Figure 3B*). About 35% are using facility management software (PPMS, iLab, Open IRIS or Agendo), 35% are using data management software (LIMS or ELN), and 4% of are using both. PPMS is the most used management software in self-service facilities, while iLab, open IRIS and Agendo dominate in full-service facilities (*Figure 3—figure supplement 3*). Notably, a quarter of facilities uses software solutions mentioned only once or twice in our survey ("other" in *Figure 3—figure supplement 3*).

However, respondents also mentioned drawbacks in using a specialized software, such as

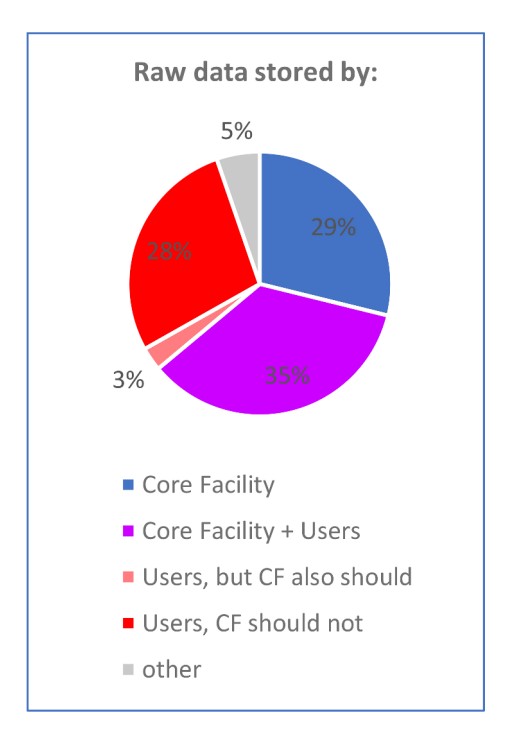

**Figure 5.** Responsibility for the long-term storage of raw data. Facilities were asked who is responsible for the long-term storage of raw data. When facilities were not responsible, they were asked if they thought they should be.

lack of cooperation of users and difficulties in customization for heterogeneous and often complex projects. The implementation and cost of such software were also considered a problem (data not shown).

We also asked facilities if they had implemented a "data management plan" instructing how research data will be annotated, stored and analyzed. Data management plans ensure that all data remain traceable, and are used in 30% of full-service facilities but only 10% of self-service facilities (*Figure 3C*). Another 50% of full-service facilities believe it would be useful, while only 20% of self-service do.

Looking into the different aspects of data management, we saw that about half of respondents had implemented data management measures to ensure that data are complete, attributable, reusable, compatible, searchable and findable (*Figure 3—figure supplement 4*).

Strikingly, 72% of all core facilities do not have a system to identify raw data used for published figures (85% and 63% for self-service and full-service respectively; *Figure 3D*). The

remaining facilities mentioned that they use a data management software (ELN, LIMS), unique IDs or a public repository to trace raw data (data not shown). The problem of non-traceable data is linked to the issue of insufficient documentation of experiments, which is clearly more pronounced in self-service facilities (*Figure 3E*). Only 20% of self-service facilities have enough documentation, while 70% of full-service facilities document their experiments sufficiently. Importantly, one half of self-service facilities does not actually know, how the experiments are documented. This might be connected to the issue of communication and responsibility addressed below.

### Communication, respect and trust between facilities and users

Communication plays a critical role in the interaction between core facilities and their users. Facilities provide a service based on their users' requirements and users need to prepare their samples and experiments according to the advice of facility staff who have expertise in the equipment and techniques available in their facility. Communication is regarded by core facilities as a sensitive issue and the interaction with the users is seen as a challenge (*Figure 2B*). About 37% of facilities feel that the communication with their users needs to be improved (*Figure 4A*). Communication between facility staff and users is mostly done through emails and/or in person, facilities say it could be improved by using a communication management software or a chat/discussion platform, and by actively motivating users to cooperate and read the information provided (data not shown).

Communication between staff and users is a common cause of tensions. The following selected comments from different respondents illustrate these tensions:

- "If I produce a plan, it will just be another formal document that will be ignored..."
- "It is very hard to get users to engage in this [quality control of data analysis]. It's hard enough to get them to use the correct controls!"
- "One of the problems that we have had is scientists thinking they know how to do analysis and using the incorrect statistical test or website because it gives them the answer they were after rather than the correct answer."
- "It is sometimes difficult – usually more because of group leaders than because of

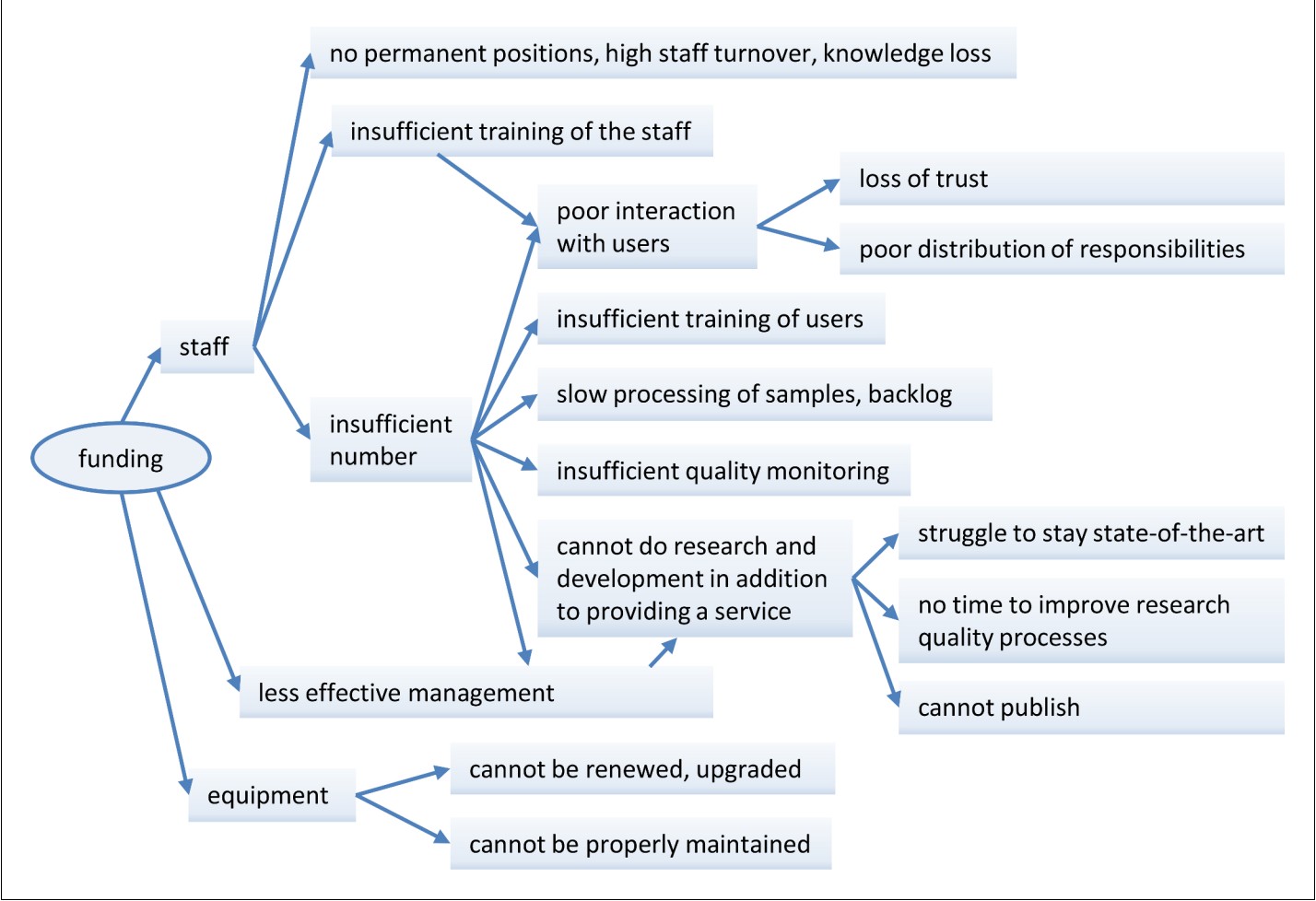

**Figure 6.** Main repercussions of insufficient funding on quality revealed by the survey. A shortage of funding (left) will have adverse impacts on staff (top), management (middle) and equipment (bottom), each of which will have further repercussions on research quality. This schematic figure shows how the different aspects of quality discussed in this paper are linked together. Additional files.

students – to get people to accept new or improved ways of doing certain types of experiments."

- "The core facility tried to implement a data management plan, but this was not accepted by the user."

These quotes also reveal another important issue: many users (or their PIs) seem not to trust or respect the expertise of facility personnel. Although core facilities are committed to help their users and most of them (85–99%) provide information from experimental design to publication (*Figure 1B*), there is a gap in the uptake from the user's side. About half of self-service facilities estimate that their users use this information only rarely or just sometimes (*Figure 4B*). Users use the provided information more frequently in hybrid-service and full-service facilities, which is likely due to the need to

conform to the facility's specific instructions for sample preparation.

Furthermore, issues with communication and trust affect another aspect critical for good quality research – the evaluation of sample quality (also discussed in the sections research quality above). Over 70% and 95% of facilities (full-service and self-service respectively) do not have a full knowledge how samples are prepared (*Figure 4C*).

Finally, less than 20% of users of self-service facilities contact the facility (always or often) before publishing their results (*Figure 4D*), whereas as facilities firmly believe that doing this would improve the quality of the published data (*Figure 2G*).

Together these results show that even though communication, trust and respect do not belong to the experimental procedure per se, they

nevertheless must be fostered as they are essential for good science.

### Sharing responsibility between facilities and users

Unexpectedly, numerous answers to the free text questions raised the issue of responsibility, although our survey did not specifically examine this aspect. The words "responsible" or "responsibility" were mentioned 123 times in total, referring to issues ranging from experimental design to publication. Notably, responses from facilities revealed an ambiguity in discerning "who is responsible for what". Most facilities do not see themselves responsible, as one respondent explained: "We allow poor samples to be processed, since the responsibility for the experiment lies entirely on the researcher! ". Other responses included: "The users are responsible for their data" and "We strongly feel that responsibility for data analysis and interpretation must be in hands of researchers, especially in the hands of research group leaders who are responsible for final research outcome."

On the other hand, a small number of core facilities do consider themselves responsible for the produced data quality. As one respondent wrote: "core facilities should be more involved in planning of the experiments and should be also responsible for the data generated in the core facility." Another one explained that "it is the overall responsibility of the facility to make sure that data are analyzed correctly. If a user decides to analyze their data, we will make sure at the publication stage that all data and conclusions drawn are consistent." The responsibility for data quality can also be integrated into the internal rules: "It is the policy of our institute that all data generated through platforms is checked by the platform staff/head before publication".

The lack of clarity in the responsibility sharing can negatively affect the quality of research. As an example, core facilities do not agree who should store the raw data, which could also be one of the reasons for the lack of traceability (as presented in the section on data management). Many respondents believe that the "long-term [data] storage is a responsibility of each individual group leader". The 30% of facilities that do not offer data storage mostly believe they should not (*Figure 5*). Yet, one respondent acknowledged the merit in storing raw data: "I think the responsibility for storing data and having back-ups is with the user. However, to have a backup of raw data at the core facility would

possibly discourage users to perform improper data manipulation and could help to solve issues on scientific misconduct."

As mentioned before, core facility staff are not listed as authors on most publications. Contradictorily, the majority of facilities believe that being part of the publication process would improve the quality of the published data, but at the same time they claim that they are not responsible for the generated data. It is important to realize that responsibility cannot be simply off-loaded. All the parties involved in the experiments share responsibility for the generated data. This especially applies to all the authors on a publication, who all share responsibility for accuracy of the published data. As one respondent wrote: "Being part of publication is holding responsibility for the work done. Neither the researchers, inclusively the PIs, are in a place to take responsibility for work with technologies that they do not understand."

## Discussion

We surveyed 253 core facilities in Europe to gain insight into their research practices and interaction with their users. Our results show that core facilities are generally invested in implementing best research practices, support transparency, rigor and reproducibility and protect against cognitive bias, which corroborates the ABRF survey's conclusions (*Knudtson et al., 2019*). The ABRF survey identified the lack of training, mentorship or oversight as the main factors contributing to the lack of compliance with rigorous and reproducible research. Similarly, respondents to our survey cited training, advising and communicating with users as the most important factors for achieving research quality. In both surveys, respondents listed mostly identical tools to improve research quality. On the other hand, the major challenges in promoting best scientific practices differed in the two surveys. While funding and lack of staff was most critical for our respondents, it was poor sample quality and lack of training in the ABRF survey (*Knudtson et al., 2019*).

Our survey reveals several weaker areas with a potential for improvement. Insufficient funding remains the major issue for the majority of core facilities. While the lack of funding can be considered a cliché, it is nevertheless connected to all aspects affecting rigor and reproducibility. It affects not only the ability to purchase and maintain state-of-the-art equipment, but perhaps even more importantly, it can directly or

indirectly affect the research quality at multiple levels (*Figure 6*). For example, the inability to hire, train and retain enough qualified staff can lead to slow processing of samples, insufficient quality monitoring and poor interaction with users, which in turn leads to loss of the users' trust and respect. Another consequence of insufficient funding can be an inefficient daily facility management, depriving the staff of the already limited time and thus preventing them to engage in other tasks such as technology development to maintain state-of-the-art techniques and publication output, which ironically can result in funding reduction in the future.

The majority of core facilities recognize the need for monitoring the quality through the whole experimental process. Based on the responses we propose that the core facilities incorporate at least the following four quality checkpoints to efficiently ensure research quality with the active help of the users (*Table 1*):

1. An experimental design check to reject any ill-designed project or improve them.
2. A sample quality control to reject poor samples. This would avoid running costly experiments unnecessarily and would ensure solid data for interpretation.
3. A data analysis check would ensure rigor and transparency and would decrease experimental bias.
4. A final check before publication would allow to make sure that the results are presented optimally and comply with best research practices.

The proposed check points need to be adjusted to the needs of each facility. For example, blinding and randomization are very important aspects of experimental design in animal core facilities. The core facilities with a large number of users might not have the capacity to perform the data analysis and publication checkpoints. In this case, the data analysis checkpoint can be assigned to experienced PIs or other qualified scientists outside of the facility (e.g. statistician, bioinformatician). This needs to be discussed and decided before starting the experiment and be part of the user agreement. The publication checkpoint is the least important, as editors and reviewers will also be involved. However, the facility should always be informed about the publications, as these are often required to secure further funding.

In addition to the above listed checkpoints, only a precise and relevant documentation can guarantee data traceability. All these aspects should be considered to achieve rigor, reproducibility and traceability. Users of self-service facilities would particularly benefit from the expertise of facility staff as most core facilities offer information to users on all stages of the research process (from experimental design to publication). This is especially relevant for techniques that are new to users (and their PIs).

Management software was rated as one of the best tools to improve research quality (which includes ensuring the traceability of the data collected). The software solutions used by core facilities are very heterogeneous with the implementation and cost being the biggest obstacles preventing wider usage. Importantly, some respondents noted that there is currently no management software allowing the full management of the facility, from booking scheduling, experimental design, data acquisition and analysis to billing. Development of such software would likely reduce the time facilities lose by switching between two or more incompatible software packages and increase the traceability of the data. Critically, an ideal management software must be user-friendly, simple and fast, as users are not willing to use an overly

**Table 1.** The proposed four checkpoints to improve quality of research in core facilities.
Based on the current situation in core facilities (CFs) revealed by the survey, four checkpoints were identified, which have the highest potential to improve rigor and reproducibility.

| Checkpoints | Recommendations |
| --- | --- |
| Experimental design | CFs should provide information and advice on the experimental design and encourage their users to follow good research practices. They should check the experimental design and reject any ill-designed project. |
| Sample quality | CFs should control sample quality before starting the experiment and reject samples of poor quality. In case of limiting or rare samples, CFs should discuss with their users what consequences the sample quality will have on data interpretation and if the experiment should continue. |
| Data analysis | CFs and PIs should decide who will be responsible for checking data analysis. |
| Publication | CFs should be informed before the data produced at the CF are submitted for publication to have the possibility to check them if they wish to. |

complicated and time demanding software and might refuse to cooperate.

Apart from the general data management described above, many core facilities require specific solutions capable of handling the particular type and amount of data they produce. For example, efficiently storing and querying large amounts of data from sequencing, microscopy or mass spectroscopy experiments each require tailored software solutions. However, the general and specialized data management systems should be interconnected and allow the attribution of the appropriate data set to each experiment or user.

Another sensitive point in the facility-user interaction turned out to be also a fundamental one: communication. Deficient communication between facility staff and their users can directly affect science quality. One third of core facilities are not satisfied with the current situation and wishes to improve the communication with their users. Communication between users and facility staff can be challenging for two reasons. First, handling questions or requests from many users on an individual basis can easily overload a facility if it is understaffed. In this regard, a tailored management software can lighten the load on the facility staff. Other suggestions from our respondents include the use of online chats or blogs, with the advantage to directly interact with multiple users at once. Secondly, tensions were frequently reported, when users ignore or do not make optimal use of the information that is provided to them by facilities. This can be the consequence of the lack of staff, which does not have enough time to communicate with the users as mentioned above. However, it can also result from facility staff having insufficient communication skills. Dealing with users with different personalities and scientific or cultural backgrounds requires good soft skills and facility staff would benefit from dedicated training in communication. In addition, core facilities should make sure that the information provided to users is clear, comprehensive, easy to follow and timely, which will encourage the user to use it.

As mentioned above, questions about who is responsible for the quality of the data collected at core facilities, and who is included as an author on papers that rely on such data, can lead to tensions between facility staff and users. The Core for Life (http://www.coreforlife.eu), an alliance of core facilities for the life sciences in Europe, has set up a working group to look at these issues (*Core for Life, 2016*). In general it is a good idea for facilities to have user agreements that cover these and other questions.

In summary, the survey highlighted issues that affect the quality of research at core facilities and could be remedied by rather simple measures. First, relevant quality checkpoints should be introduced at sample submission, after data analysis and also just before publication. Second, data management should be further improved in most core facilities, and the use of management software would be beneficial. Third, there is a need for improvement of the communication between facility staff and users, which requires tact and effort and is often perceived as a challenge. In addition, the responsibilities of each party should be clearly defined. The survey reported here is part of the Q-CoFa (Quality in Core Facilities) project that aims to develop a framework for the best quality practices at the interface between core facilities and researchers and provide guidance on communication, information flow and data management to ensure the generation of rigorous and reproducible data. We are working on guidelines which we will publish to further strengthen the role of core facilities to increase and promote research data quality.

## Materials and methods

We developed a 68-question online survey using LimeSurvey software. We initially aimed to reach to all core facilities in Europe. We used the Google search engine in English language with the keywords "core facility" in 19 countries. Subsequently we also visited the websites of the major Universities in each country. We stopped after retrieving 1000 email addresses, as further searching retrieved only a limited number of websites in local languages lacking English translation. The leaders of these facilities were then contacted by email. In addition, our survey was publicized in the CTLS newsletter (Core Technologies for Life Sciences) and several facilities we contacted by initial email further forwarded the survey link to their colleagues. The survey was open from December 2019 to July 2020. All respondents were anonymous. We received 276 total forms (28% participation rate), 253 of which were complete. These numbers do not include the four respondents that did not give us permission to publish their results. To estimate the margin of error in our survey, we used the publicly available sample size calculator (https://www.surveysystem.com/sscalc.htm). Assuming an equal (1:1) answer distribution (the worst-

case scenario), the sample size of 253 respondents from 1000 core facilities (population) corresponds to a 5.3% margin of error at 95% confidence level.

The survey contained yes/no, multiple-choice and open-field text questions. The survey data was analyzed using Microsoft Office 365 Excel. We had 28 free text fields to allow the respondents express themselves freely, to eliminate potential bias stemming from suggested answers. Open-field answers were evaluated by reading each of them personally and defining categories manually based on the replies so that they correspond to the opinions of the respondents as faithfully as possible. Keywords were then chosen to allow automatic counting in Excel. The survey questions are in *Supplementary file 1*.

We analyzed the data in three different ways: (1) all facilities together, (2) facilities grouped by their type/specialization (genomics, microscopy, etc) and (3) grouped by their operating mode (full-, hybrid-, self-service). While grouping the core facilities by type showed differences, these were often too specific to each type and could not be generalized with respect to quality procedures. In addition, some groups were too small to allow conclusive statements. On the other hand, grouping the facilities by operating mode revealed clear and meaningful differences between the groups in their approach to quality procedures. Therefore, the manuscript presents results from either all facilities together or grouped by their operating mode. All charts with all three groupings of data are included in the Excel file, containing the raw and analyzed data, which are available on Dryad doi:10.5061/dryad.zkh18938m.

### Limitations

1000 core facilities were invited to participate in the survey and only a quarter completed the survey. It is possible that facilities with concerns about research quality were more likely to participate in the survey, therefore causing a selection bias. Additionally, the survey targeted only facility staff and thus lacks the users' point of view.

### Acknowledgements

We thank Dr Martin Kos and Dr Nicolas Sylvius for helpful discussions and critical reading of the manuscript. We thank Dr Anton Bespalov and Dr Christoph Emmerich (PAASP) and Dr Barbara Hendriks (DZHW) for their help designing the survey, valuable discussions and critical reading of the manuscript. We thank Dr Charles Girardot for helpful discussions about management software. We also thank all the respondents for their time. This work was supported by the BMBF and performed at the Interdisciplinary Neurobehavioral Core (INBC) at the University of Heidelberg (RRID:SCR_019153).

**Isabelle C Kos-Braun** is in the Interdisciplinary Neurobehavioral Core, Heidelberg University, Heidelberg, Germany

isabelle.kos@pharma.uni-heidelberg.de

https://orcid.org/0000-0002-2380-5720

**Björn Gerlach** is at PAASP GmbH, Heidelberg, Germany

https://orcid.org/0000-0002-4900-6302

**Claudia Pitzer** is in the Interdisciplinary Neurobehavioral Core, Heidelberg University, Heidelberg, Germany

Claudia.pitzer@pharma.uni-heidelberg.de

*Author contributions:* Isabelle C Kos-Braun, Conceptualization, Data curation, Formal analysis, Investigation, Visualization, Methodology, Writing - original draft, Writing - review and editing; Björn Gerlach, Conceptualization, Formal analysis, Validation, Methodology, Writing - review and editing; Claudia Pitzer, Conceptualization, Resources, Supervision, Funding acquisition, Validation, Methodology, Project administration, Writing - review and editing

*Competing interests:* The authors declare that no competing interests exist.

### Funding

| Funder | Grant reference number | Author |
|---|---|---|
| Bundesministerium für Bildung und Forschung | 01PW18001 | Isabelle C Kos-Braun Björn Gerlach Claudia Pitzer |

The funders had no role in study design, data collection and interpretation, or the decision to submit the work for publication.

### Decision letter and Author response

Decision letter https://doi.org/10.7554/eLife.62212.sa1
Author response https://doi.org/10.7554/eLife.62212.sa2

## Additional files

### Supplementary files

• Supplementary file 1. Survey questionnaire.

- Supplementary file 2. Excel spreadsheet containing responses to questionnaire.
- Transparent reporting form

## Data availability

The source data and analysed data have been deposited in Dryad under the accession code https://doi.org/10.5061/dryad.zkh18938m.

The following dataset was generated:

| Author(s) | Year | Dataset URL | Database and Identifier |
|---|---|---|---|
| Kos-Braun I, Gerlach B, Pitzer C | 2020 | https://doi.org/10.5061/dryad.zkh18938m | Dryad Digital Repository, 10.5061/dryad.zkh18938m |

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
