## [Decision Letter]

Thank you for submitting your article "Survey of Core Facilities shows the importance of communication and management for optimal research quality" to *eLife* for consideration as a Feature Article. Your article has been reviewed by three peer reviewers, and the evaluation has been overseen the *eLife* Features Editor, Peter Rodgers. The following individuals involved in review of your submission have agreed to reveal their identity: Guillermo Marques (Reviewer #1); Sven Nahnsen (Reviewer #2); Daniele Soroldoni (Reviewer #3).

The reviewers and editors have discussed the reviews and we have drafted this decision letter to help you prepare a revised submission.

Summary:

This is a solid study on an important topic. It is broader and significantly different from the survey of Knudtson (2019), and uncovers a different but largely overlapping set of issues. It brings to the front and quantifies a known problem: research resources are squandered and rigor and reproducibility suffer because poor communication and distrust between personnel at central facilities (CFs) and the clients of CFs. The study also identifies the issue of shared responsibility as a critical aspect to address to improve the quality of the research generated in CF. The survey is comprehensive and the analysis does not overreach in the interpretation of the data. The proposed checkpoints are also logical but they will introduce delay and won't scale. There are also a number of points and concerns about the work that need to be addressed to make the article suitable for publication.

Essential revisions:

1) Please explain why the survey was aimed at the personnel of central facilities (CFs), and not at other stakeholders, notably the users of CFs (see below for more on this), and also the senior management of institutions, who may have certain expectations on the CF performance.

2) I would like a more nuanced discussion on the relationship between CF personnel and clients. A critical point uncovered by this study is that, in the view of CF personnel, clients do not accord them enough scientific respect, and oftentimes ignore the CF recommendations. While this is certainly the case in my experience, a bit more soul searching on the part of CF personnel could uncover some causes for this behavior that fall on the CF's side, and not the clients'. Case in point: poor communications by CF staff with clients may have something to do with clients ignoring advice. Similarly, the lack of enough qualified staff (Figure 2B) may have a bearing on this.

3) Unsurprisingly, funding (or lack thereof) is identified as the biggest challenge to better CF quality (Figure 2B). However the chain of undesirable events triggered by inadequate funding does not include poor interaction with users (Figure S4), that is identified as a major challenge to achieve research quality (2B). I think there is a clear link between the two, and that is the lack of enough qualified staff in the absence of adequate funding (Figure 2A).

4) The Discussion does a good job of highlighting the critical findings and proposing improvements, but I miss a better effort to paint a coherent picture of the different findings and how they are, in my view, inextricably linked. The issues brought up in point 2 above (poor communications, inadequate staff) make it difficult to achieve a level of reciprocal trust with investigators that allows shared responsibilities and full integration of CF staff in the client's research. Inadequate funding prevents the hiring, development and retention of qualified personnel. I find the closing sentence in the Introduction is wishful thinking: substantial funding will be needed to bring already stressed CF to the staffing levels needed (in quantity and quality) to be able to provide the desired scientific partnership with clients.

5) The discussion of data management could be improved. Software like LIMS and ELN can help with some data-management tasks, but they would not be able to help with a task such as querying petabytes of sequencing data in order to prepare a figure. Please say more about the different kinds of data-management tasks that need to be performed at a CF.

6) A key characteristic of any core facility is cost recovery (eg through user fees). Please explain why this topic was not included in the survey.

---

## [Author Response]

Essential revisions:1) Please explain why the survey was aimed at the personnel of central facilities (CFs), and not at other stakeholders, notably the users of CFs (see below for more on this), and also the senior management of institutions, who may have certain expectations on the CF performance.

We agree that it would have been interesting to have the users’ opinion as well, however, apart from the pressure due to time-limited funding, there are two major reasons why we did not include the users in our survey.

1) In order to be permitted to perform the current survey, we had to comply with the data privacy regulations to keep the respondents anonymous. This turned out to be more complicated than we expected. Creating a survey for users of each facility while maintaining anonymity of both the users and CFs’ staff was not logistically possible without introducing a bias (see point 2).

2) We did not find a feasible and affordable way to include the users without introducing a bias:

a) If we had asked the CFs to forward our survey to their users, we would have had no control over the bias due to the selection of the users contacted by the facility.

b) On the other hand, we also could not ask each facility to give us a list of their users in order to contact them ourselves and at the same time keep the users anonymous.

c) In addition, we would receive different number of responses from users of each facility, which we could not normalize due to the already mentioned data privacy regulations.

Apart from the reasons mentioned above, this survey is a part of a larger project, aiming to develop recommendations for best scientific practices in core facilities. As such the survey was the starting point to get a general insight about the current situation at CFs and this is why it was targeted at CFs leaders. Therefore, we did not consider at this stage to include senior management, which might have expectation but cannot answer question about the daily functioning of the facility.

2) I would like a more nuanced discussion on the relationship between CF personnel and clients. A critical point uncovered by this study is that, in the view of CF personnel, clients do not accord them enough scientific respect, and oftentimes ignore the CF recommendations. While this is certainly the case in my experience, a bit more soul searching on the part of CF personnel could uncover some causes for this behavior that fall on the CF's side, and not the clients'. Case in point: poor communications by CF staff with clients may have something to do with clients ignoring advice. Similarly, the lack of enough qualified staff (Figure 2B) may have a bearing on this.

Thank you for your suggestion. We agree that our original discussion of this issue was not sufficient. We restructured and expanded the paragraph about communication in the Discussion and included these two points as well. We also made clear that software cannot change the users’ reluctance to accept advice from CFs.

The modified paragraph starts with: “Another sensitive point in the CF-user interaction turned out to be also a fundamental one: communication. Deficient communication between CF staff and their users can directly affect science quality…”

3) Unsurprisingly, funding (or lack thereof) is identified as the biggest challenge to better CF quality (Figure 2B). However the chain of undesirable events triggered by inadequate funding does not include poor interaction with users (Figure S4), that is identified as a major challenge to achieve research quality (2B). I think there is a clear link between the two, and that is the lack of enough qualified staff in the absence of adequate funding (Figure 2A).

Thank you for pointing this out. We added the missing link in the Figure S4 (now Figure 6, please see next comment No. 4) and also included the effect of insufficient funding on the interaction with the users in the corresponding paragraph in the Discussion. “For example, the inability to hire, train and retain enough qualified staff can lead to slow processing of samples, insufficient quality monitoring and poor interaction with users, which in turn leads to loss of the users’ trust and respect.”

4) The Discussion does a good job of highlighting the critical findings and proposing improvements, but I miss a better effort to paint a coherent picture of the different findings and how they are, in my view, inextricably linked. The issues brought up in point 2 above (poor communications, inadequate staff) make it difficult to achieve a level of reciprocal trust with investigators that allows shared responsibilities and full integration of CF staff in the client's research. Inadequate funding prevents the hiring, development and retention of qualified personnel. I find the closing sentence in the Introduction is wishful thinking: substantial funding will be needed to bring already stressed CF to the staffing levels needed (in quantity and quality) to be able to provide the desired scientific partnership with clients.

This is a good suggestion. We expanded the paragraph about the lack of funding and its repercussions on the different aspects of research quality.

“Insufficient funding remains the major issue for the majority of CFs […]”

We discuss how poor communication and loss of reciprocal trust can be in fact an indirect consequence of inadequate funding. Furthermore, the previous supplementary figure 4 was improved to illustrate better the connections between the different aspects influencing quality and is now included as a main Figure 6.

The last sentence of the Introduction was removed.

5) The discussion of data management could be improved. Software like LIMS and ELN can help with some data-management tasks, but they would not be able to help with a task such as querying petabytes of sequencing data in order to prepare a figure. Please say more about the different kinds of data-management tasks that need to be performed at a CF.

Thank you for your suggestion, we added a paragraph discussing the need for specialized data management solutions by individual CFs.

The following new paragraph was added in the Discussion: “Apart from the general data management described above, many CFs require specific solutions […]”

6) A key characteristic of any core facility is cost recovery (eg through user fees). Please explain why this topic was not included in the survey.

We agree that the cost recovery is an important issue for any CF and that funding can directly affect quality. Nevertheless, we did not include it in this survey because we wanted to focus on the aspects affecting the science quality regardless of the cost. Importantly, many CFs are established to provide affordable access to the state-of-the-art technology. Many CFs rely mainly on external funding because their user fees cannot cover the cost. Often the user fees are not even meant to cover the cost, but only to participate to it. Moreover, the funding schemes of CFs differ wildly, even within one country. We do not feel that covering this topic would be beneficial for our initial analysis.